# Protocol for a QUALICOPC study in a Malaysian setting: Primary care pharmacy services system performance and T2DM older adults' quality of life evaluation

Nurain Suleiman[1,2], Ishtiaq Ahmad[1]*, Motoyuki Yuasa[1]

1 Department of Global Health Research, Graduate School of Medicine, Juntendo University, Tokyo, Japan, 2 Ministry of Health Malaysia, Training Division, Federal Government Administrative Centre, Putrajaya, Malaysia

* ahmad@juntendo.ac.jp

## Abstract

Pharmacists may play a crucial role in enhancing the primary care (PC) system by offering convenient patient care and leveraging their expertise to promote community health through pharmaceutical care and value-added services. Given the significant healthcare issue of Type 2 Diabetes Mellitus (T2DM) in Malaysia and other Asian countries, the associated treatment costs have become a global economic concern. The increasing prevalence of T2DM in Malaysia, particularly among older adults, is intensifying its clinical and economic impact on the country. To address this, the current study aims to evaluate the effectiveness of PC pharmacies using the Quality and Costs of Primary Care (QUALICOPC) Survey, which is widely used in Europe and other developed and developing countries. This research seeks to collect experience and value feedback from older adults with T2DM or their caregivers, and job satisfaction feedback from pharmacy staff. Additionally, the quality of life (QOL) of older adults with T2DM will be assessed among the patients who are respondents in this study. This study will use the adapted and validated QUALICOPC questionnaires that are in line with international standards and tailored to the Malaysian context. An expert panel, including the Pharmaceutical Services Division of the Ministry of Health (MOH) Malaysia and the Kulliyyah of Pharmacy at the International Islamic University Malaysia (IIUM), has been involved in the content validity of the adapted QUALICOPC questionnaires. The QUALICOPC questionnaires will be validated for the construct with a minimum of 38 samples for each client experience questionnaire, 30 samples for each value questionnaire, and 19 samples for the staff questionnaire. The survey will involve 192 samples from each group of T2DM older adults or caregivers and all corresponding public PC pharmacy staff. The QOL will be evaluated using the EuroQol (EQ) 5-dimensions 5-levels [EQ-5D-5L] and an EQ visual analog scale (EQ-VAS). The findings from this study will be valuable for informing the World

**Data availability statement:** No datasets were generated or analysed during the current study. All relevant data from this study will be made available upon study completion.

**Funding:** Nurain Suleiman received research support allowance from Japan International Cooperation Agency (JICA) for conducting this research. A grant also is currently being applied as financial aid for this research. The funders had no role in study design, data collection and analysis, decision to publish, or preparation of the manuscript.

**Competing interests:** The authors have declared that no competing interests exist.

Health Organization (WHO) and the Malaysian government in designing more effective health policies.

## Introduction

The current diabetes prevalence in Asian nations is concerning, with around 60% of the population diagnosed with the condition [1]. Patient medication non-adherence (PMNA) can worsen diabetes complications, and the increasing number of newly diagnosed diabetes cases in Malaysia as well as other countries has the potential to exaggerate the current epidemiological situation [1,2]. PMNA is a significant issue in global healthcare [2–5], as it can result in unwanted health outcomes for individuals with chronic conditions [2,5]. Improved PMNA in diabetes patients may enhance the effectiveness of oral antidiabetic (OAD) medications, leading to better blood sugar control and potential glycaemic improvement [6].

Pharmacists may enhance primary care (PC) pharmacy services by improving PMNA significantly. The International Pharmaceutical Federation (FIP) and World Health Organization (WHO) recommend that pharmacists be recognised as healthcare professionals, not only for consulting on health-related issues but also for providing healthcare services in a community setting [7]. Developing nations such as Malaysia, Indonesia, and Nepal have expanded the roles of pharmacists and improved pharmacy services provided by the PC pharmacy system to meet the pharmaceutical care needs of the community [8–10].

Health system goals, including improving health, providing financial security, and ensuring citizen satisfaction, are crucial for optimal health system performance [11]. In Malaysia, an extensive network of public health clinics with integrated pharmacies provides comprehensive and affordable treatment to most of the population, regardless of their urban or rural location. Therefore, the performance of public pharmacy departments can significantly impact the overall effectiveness of the country's healthcare system.

The Quality and Costs of Primary Care (QUALICOPC) Survey is a critical measure of clinical quality that reflects the effectiveness of care. This survey has been used in over 30 countries worldwide, primarily in developed countries (such as New Zealand and European countries), as well as in developing countries (such as Turkey and Malaysia), to examine and compare PC services system performance [12] since it combines insights from both wealthy and poor nations, thus providing a comprehensive view of health improvement [13].

In Malaysia, diabetes prevalence is increasing with age, ranging from 5.5% to 39.1% for the 18–19 and 70–74 years of age groups, respectively [14]. Additionally, there is a lack of evaluation of pharmacy services related to the quality of life (QOL) of older adults with Type 2 Diabetes Mellitus (T2DM) in Malaysia. Thus, the researcher has included an evaluation of patients' QOL in this study. The previous international QUALICOPC survey did not include patient health outcomes [12]. There was limited research in the field of PC examining the relationship between

multimorbidity and health-related quality of life (HRQOL), and this issue was not only observed in Malaysia [15]. Therefore, conducting thorough studies on affected patients is essential to improve treatment strategies [15].

Patient satisfaction significantly impacts individual's QOL [6] and medication adherence in diabetes treatment [16]. The QUALICOPC Survey of primary care pharmacy services system performance (PCSSP) and the assessment of patients' QOL can help identify the strengths and weaknesses of the PC pharmacy in both developed and developing countries. This can help to improve pharmaceutical needs and reduce disparities in managing chronic conditions like diabetes mellitus (DM) in PC pharmacy settings. Also, studying other countries' experiences, especially those in Europe with effective PC systems, can provide valuable benchmarking insights for this purpose [12].

## Importance of QUALICOPC study for strengthening Primary care (PC) pharmacy

Primary care (PC), also referred to as generalist care, acts as the first access point to the healthcare system, offering easily accessible services to the general population, irrespective of their health conditions, and is conveniently located near patients' residences [17]. The effectiveness of a PC system relies on integrating PC functions at both the structure and service provision levels within the healthcare system [17].

The PC system in Malaysia encompasses both public and private sectors. Private healthcare facilities in urban areas provide medical and diagnostic services, delivered by private practitioners, including doctors, physicians, dental clinics, and retail pharmacies [18]. In contrast, the Ministry of Health (MOH) in Malaysia has established an extensive network of public PCs [18]. The public PC health clinics range from large urban facilities (Type 1–3) overseen by family medicine specialists and offering supplementary services such as pharmacy, laboratory, and radiology support, to smaller clinics (Type 4–6) located in rural areas and managed by medical assistants [11]. Type 7 encompasses the smallest health clinic [11].

PC Pharmacy services for chronic diseases may differ across different countries. Previously, conventional pharmacy services for chronic disease such as diabetes mellitus (DM) primarily involved medication dispensing. However, these services have evolved to encompass a wider range of pharmacist responsibilities, with a focus on providing patient-centered care [19]. Desselle et al. (2019) outlined various types of pharmacy services in a community setting, as shown in Table 1 [20].

In a Malaysian public PC pharmacy setting, the main services offered for older adult diabetes patients include pharmaceutical care and Pharmacy Value Added Services (VAS). The offered pharmaceutical care is recognised as Diabetes Medication Therapy Adherence Clinics (DMTAC). Meanwhile, some examples of Pharmacy VAS include Drive through a pharmacy, Medicine Supply via Courier or Postal Service (UMP), SMS Take&Go, Pharmacy appointment system (STJ), and Medication locker, commonly referred to as Medibox [21].

**Table 1. Several types of pharmacy services in a community setting.**

| Types of Pharmacy Services | Examples |
|---|---|
| (1) Pharmaceutical care | Clinical pharmacy services such as traditional medication therapy management; MTM (comprehensive medication review, medication action plan, personalized medication record, intervention, and referral, as well as documentation and follow-up) and continuous medication management; CoMM (pharmacists systematically review the patient's medication record and monitor every medication being dispensed to prevent, identify, and resolve drug therapy problems or obstacles to optimal therapy during the dispensing process) |
| (2) Value-added | Appointment-based models (ABM) |
| (3) Prevention and wellness | Wellness/ health promotion programs (e.g.,: smoking cessation counselling, travel health-related medications, hormonal contraception prescribing, naloxone dispensing, and education) |
| (4) Others (i.e., Monitoring/ screening services) | Lab monitoring and screening (e.g.,: anticoagulation services, osteoporosis screenings, lipid screenings, point of care conditions such as influenza, strep throat, and hepatitis C screenings) |

The DMTAC was introduced by the Ministry of Health (MOH), Malaysia in 2006 [8]. This service involves a partnership between pharmacists and physicians to enhance outpatient care services for diabetes management [8]. The DMTAC service requires patients to participate in a minimum of four (4) appointments, during which the pharmacist performs assessments of medication adherence, evaluates medication knowledge, identifies and manages drug-related problems (DRPs), provides medication counselling, offers diabetes education, and monitors clinical indicators including haemoglobin A1c (HbA1c), fasting blood sugar (FBS), and random blood sugar (RBS) [8].

The Pharmacy VAS in Malaysia provides a range of supply methods using information technology and modern applications to facilitate dispensing schedules for refilling medications between the public pharmacy department and the patient [21]. The first medication collection following a doctor's appointment should be collected from the conventional pharmacy counter to confirm any medication management alterations and to offer suitable counselling for patients [21].

The QUALICOPC aims to evaluate the performance of PC systems in Europe, focusing on quality, equity, and costs [17]. Data collection for the QUALICOPC study includes system, practice, and patient approaches [17]. The QUALICOPC Survey integrates various information sources to conduct comprehensive analyses, advancing the state of the art in PC studies and contributing to evidence-based health policy development [17].

This paper sought to conduct a more in-depth exploration of the quality of public PCPSSP in a Malaysian setting as assessed by the experience of clients (could be either T2DM older adults or their caregivers), besides exploring some aspects of healthcare delivery that the clients valued. Comparisons will also be made between two types of major services provided by the public PC pharmacy (i.e., pharmaceutical care a.k.a. DMTAC and Pharmacy VAS) as we hypothesize experiences will differ across these patient sub-groups. Pharmacy workforce composition and size together with the pharmacy staff workload will be assessed with their job satisfaction. The secondary objective includes assessing the QOL of older adults with T2DM who are the respondents in this study and receiving the main pharmacy services in Malaysia's public PC pharmacy system, which is potentially useful in both the clinical encounter and quality improvement. Finally, this paper will provide possible solutions to inform healthcare reform activities moving forward.

## Hypothesis

Data gathered from the system (structure), service provision, and user of services levels can provide valuable insights into the strengths and areas for improvement of pharmacy services, specifically DMTAC and Pharmacy VAS, within a public PC pharmacy system in a Malaysian setting. The outcomes from QUALICOPC-PCPSSP questionnaires for clients and the health-related quality of life (HRQOL) questionnaire are dependent on the DMTAC enrolment status group and Pharmacy VAS enrolment status group (Fig 1). Outcomes from the QUALICOPC-PCPSSP questionnaire for staff are dependent on the types of pharmacy occupation, while perception of staff work satisfaction is dependent on staff workload (Fig 1).

The conceptual framework outlines the approach to achieve the objectives of the study (Fig 2). This study will evaluate the performance (in terms of power and effects) which measures the system level by the existing international studies, and the users of services level by using client questionnaires to determine pharmacy accessibility, client experience, and client value (Fig 2). This study will also assess the quality of life (QOL) of older adults using the EQ-5D-5L questionnaire (Fig 2). The staff workload and the perception of staff work satisfaction will be evaluated using the staff questionnaire to determine its influence on the types of pharmacy occupation and the staff workload (Fig 2).

## Methodology

This study will employ various data collection methods across three (3) main levels of the QUALICOPC study: system level, service provision level, and users of services level (Fig 4). This research focusses on identifying 'good practices' in PC pharmacy provision, covering all aspects of PC pharmacy. Two (2) main themes are involved at the system level and three (3) main themes at the service provision and users of services levels. The research team consists of experienced researchers with expertise in global health research, pharmacy practice, and/or public health.

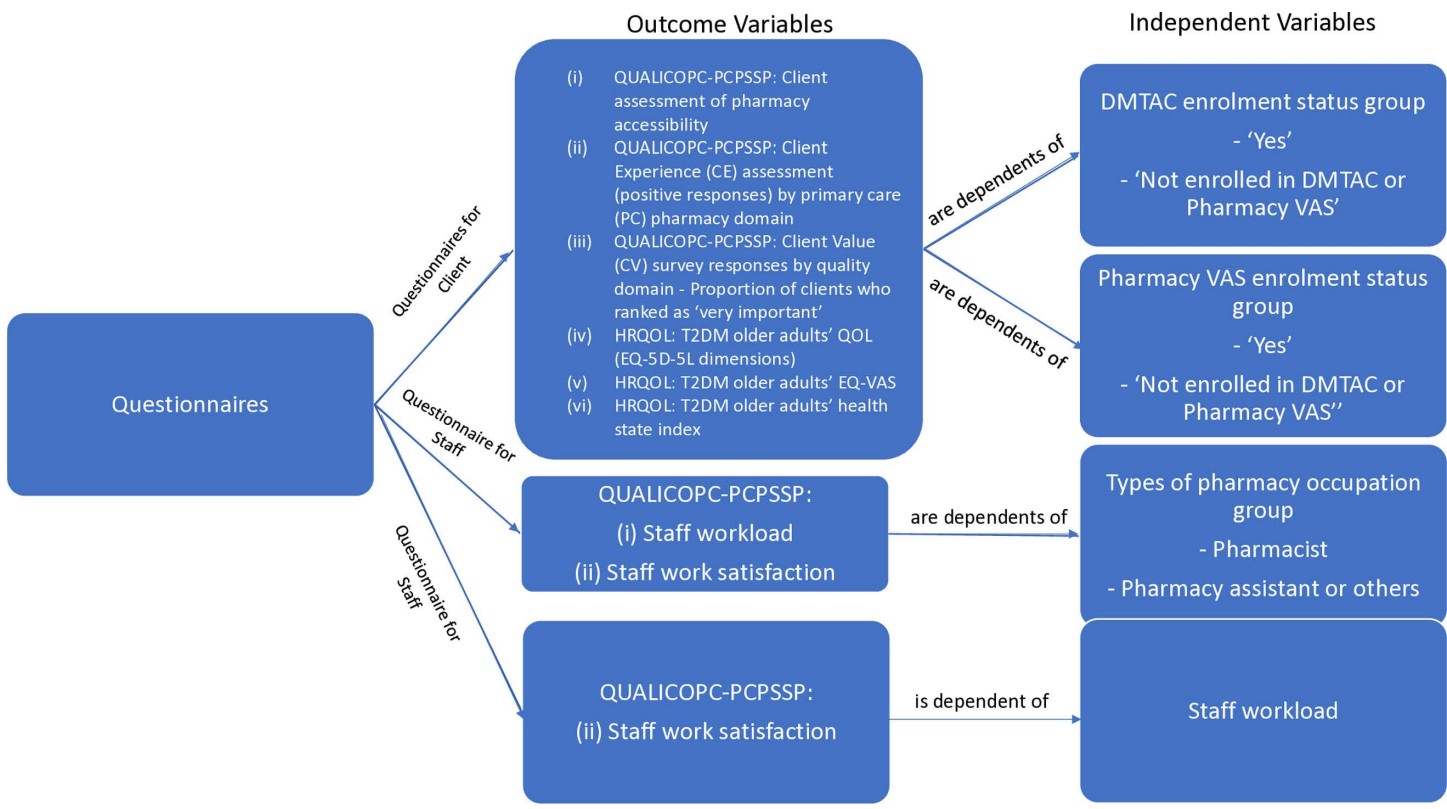

**Fig 1. Hypothesis for Questionnaires for Client and Staff.**

## (A) For System Level

This research applied easily accessible data and existing international studies to collect evidence on the structure of PC pharmacy. Demoz et al. (2018) explored the evolving roles of pharmacists in providing clinical services for diabetes care [19]. They emphasised the effects on aspects such as managing DRPs, promoting medication adherence, offering counselling and patient education, monitoring HbA1c, blood pressure, and lipid profiles, addressing diabetes-related complications, improving HRQOL, delivering economic benefits, as well as providing MTM services. [19]. Desselle et. al. (2019) identified effective strategies for implementing VAS within the PC pharmacy system [20], as outlined in S1 Table.

The two previous studies investigated two (2) primary topics: (i) the cost of PC pharmacy (concerning care efficiency), and (ii) avoidable hospitalisation (as a measure of the quality of PC pharmacy).

The author highlights the lack of patient HRQOL and economic benefits evaluation of pharmacists' roles in providing clinical services for diabetes care in a Malaysian public PC pharmacy. Furthermore, the author addressed that public PC pharmacies in a Malaysian setting are in the strategic planning in which a pharmacy has successfully implemented VAS, with infrastructure and processes in place for sustainability, thus suggesting partnering with providers for diabetes management services. This strategy may be evaluated by clinical and humanistic outcomes [20]. Desselle et. al. (2019) emphasised that when clinical and humanistic outcomes are achieved, financial outcomes also align with them [20]. Thus, the author uses the HRQOL questionnaire to measure clinical outcomes and QUALICOPC questionnaires to measure human satisfaction or humanistic outcomes.

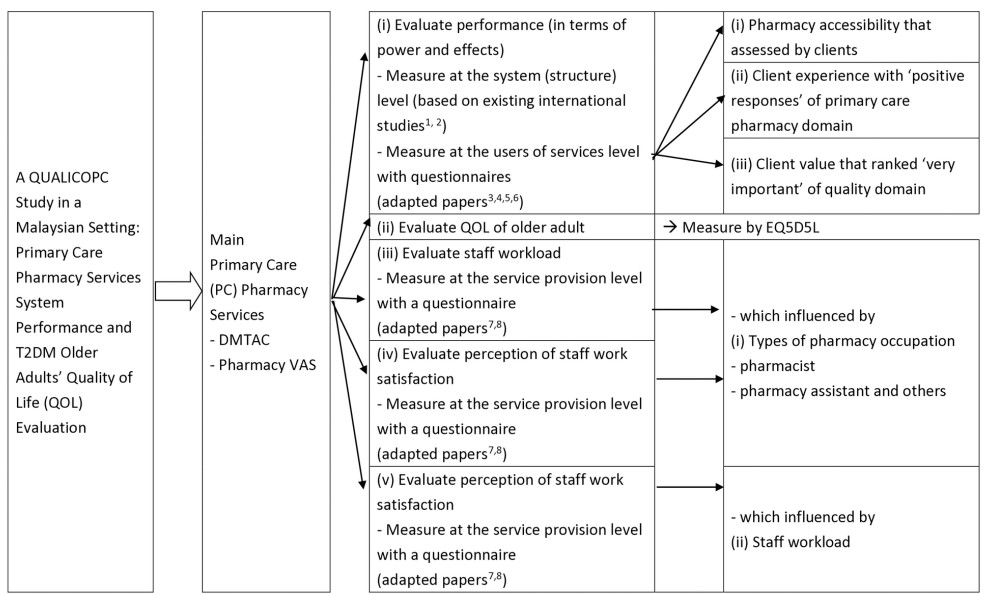

References:[1] Demoz GT, Gebremariam G, Legesse Y. The current scenario of clinical pharmacy service in the management of patients with diabetes mellitus and related complications: Review article. In J Pharm Pract. 2018;10(4):237–47. [2] Desselle SP, Moczygemba LR, Coe AB, Hess K, Zgarrick DP. Applying contemporary management principles to implementing and evaluating value-added pharmacist services. Pharmacy. 2019;7(3):99. [3] Chan HK, Shahabudin NA, Ghani NA, Hassali MA. Satisfaction with traditional counter versus value-added services for prescription claims in a Malaysian tertiary hospital. J Pharm Health Serv Res. 2015;6(1):61–8. [4] Lionis C, Papadakis S, Tatsi C, Bertsias A, Duijker G, Mekouris PB, et al. Informing primary care reform in Greece: Patient expectations and experiences (the QUALICOPC study). BMC Health Serv Res. 2017;17(1):255. [5] Schäfer W La, Boerma WG, Kringos DS, De Maeseneer J, Greß S, Heinemann S, et al. QUALICOPC, a multi-country study evaluating quality, costs and equity in primary care. BMC Fam Pract. 2011;12:115. [6] Schäfer WL, Boerma WG, Kringos DS, De Ryck E, Greß S, Heinemann S, et al. Measures of quality, costs, and equity in primary health care instruments developed to analyse and compare primary care in 35 countries. Qual Prim Care 2013;21(2), 67–79. [7] Ab Rahman, N., Husin, M., Dahian, K., Mohamad Noh, k., Atun, R., & Sivasampu, S. (2019). Job satisfaction of public and private primary care physicians in Malaysia: analysis of findings from QUALICO-PC. Human resources for health, 17(1), 1-10. [8] Hoffmann, K., Wojczewski, S., George, A., Schäfer, W. L., & Maier, M. (2015). Stressed and overworked? A cross-sectional study of the working situation of urban and rural general practitioners in Austria in the framework of the QUALICOPC project. Croatian medical journal, 56(4), 366-374.

**Fig 2. Conceptual framework of this study.**

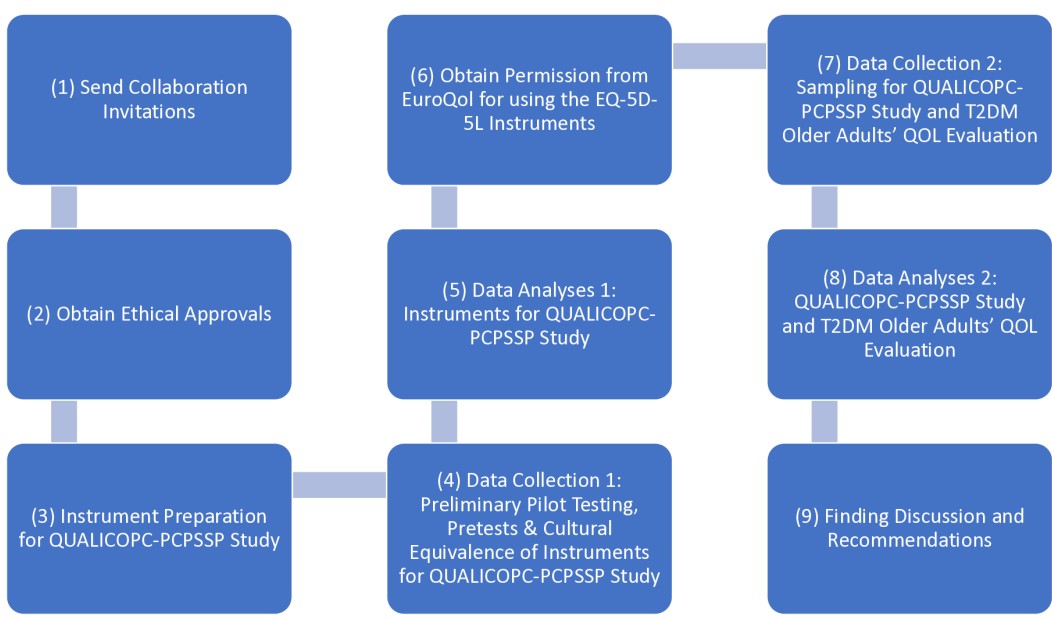

**Fig 3. Flowchart of this study. The research processes involve collaboration invitations, ethics approval, instrument preparation, phase 1 data collection and analysis, obtaining EuroQol permission, phase 2 data collection and analysis, and finding discussion and recommendation (Fig 3).**

| LEVELS | PRIMARY CARE PHARMACY FEATURES | SYSTEM GOALS (direct and indirectly contribute to health) |
|---|---|---|
| System level (PC pharmacy system) | Design and organization of primary care pharmacy<br>- Financing<br>- Regulation<br>- Resources | Access/ equity |
| Service provision level (delivery of care process at primary care pharmacy level: core providers of primary care pharmacy) | Tasks and activities<br>- First contact care<br>- Scope of service package<br>- Continuity of care<br>- Integrated provision<br>- Community orientation | Costs/ efficiency<br><br>Process quality of services |
| Users of services level (process and outcome level) | Responsiveness<br>- Accessibility<br>- Equity<br>- Convenience of services | Perceived quality of services |

**Fig. 4. Mechanism of inter-relations between the different levels and their features**[a,b]. [a]**For primary care (PC) pharmacy that leads to health outcomes.** [b]**Adapted from Schäfer et. al. (2011).**

(B) **For Service Provision Level and Users of Services Level**

**Study design.** A cross-sectional survey will be conducted over a minimum period of two (2) months.

**Study population.** Older adults (aged 60 and above) with T2DM and/or their caregivers who visited public PC pharmacies (Type 1–3) and the pharmacy staff in Malaysia.

**Study site for instrument preparation in the QUALICOPC-PCPSSP study.** Public Health Clinics in Negeri Sembilan, Malaysia, excluding Port Dickson Public Health Clinic.

**Study site for QUALICOPC-PCPSSP study and QOL evaluation.** Port Dickson Public Health Clinic, Negeri Sembilan, Malaysia

**Instruments preparation for QUALICOPC-PCPSSP study: General information.** The QUALICOPC instruments in English will be adapted, validated, and finally translated into Malay. The researcher will include questions in the survey based on previously published international studies, ensuring that all key topics for analysis are covered and suitable for use in global surveys, considering the diversity of healthcare systems in various countries [12].

For the QUALICPC-PCPSSP questionnaires are appropriate for use in a Malaysian setting, an expert panel group has been established in collaboration with the Pharmaceutical Services Division, MOH, Malaysia, and Kulliyyah of Pharmacy, International Islamic University Malaysia (IIUM) for content validity purposes. The adapted QUALICOPC questionnaires will be validated for the construct with a minimum of 38 samples for each client experience questionnaire, 30 samples for each value questionnaire, and 19 samples for the staff questionnaire. A content review will be conducted for the developed data collection form of the adapted instruments. The client questionnaires will be content reviewed based on

demographics, health status, frequency of visits, reason for current visits, and pharmacy accessibility (Fig 5). The content validity will focus on PC pharmacy (client experience) and quality (client value), with five (5) factors focusing on accessibility, continuity and coordination, communication and patient-centered care, comprehensiveness and patient activation (Fig 5). The staff questionnaire will be content reviewed on the practice workforce's composition and size (pharmacy characteristics), staff demographics, and workload, with the content validity focusing on the perception of staff work satisfaction(Fig 5).

The steps involved in validation include: (i) validation of the adapted versions, (ii) translation process, and (iii) validation of the translated versions (Fig 6). The data collection started on 18th March 2024 and continued until 31st May 2025. Validation has been conducted since 18th March 2024 followed by the actual study, which is scheduled in April 2025.

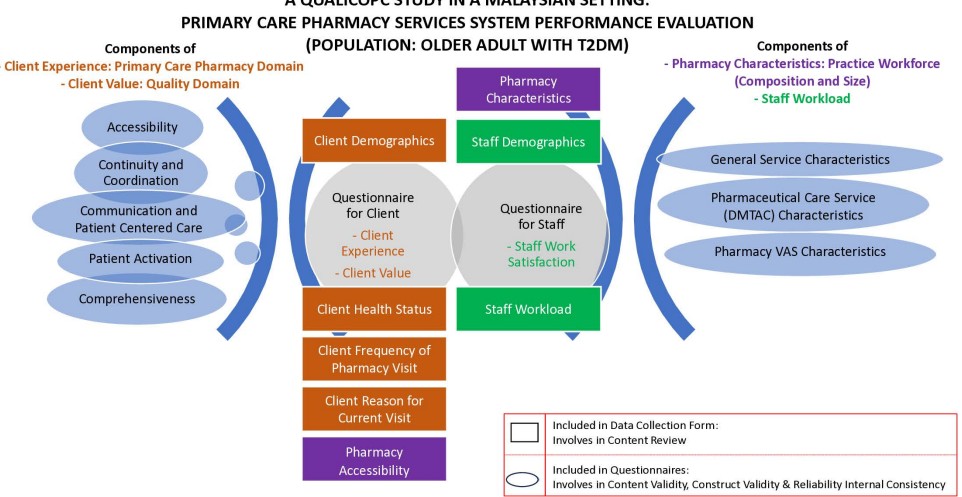

**Fig 5. Conceptual framework for instruments preparation for the QUALICOPC-PCPSSP study.**

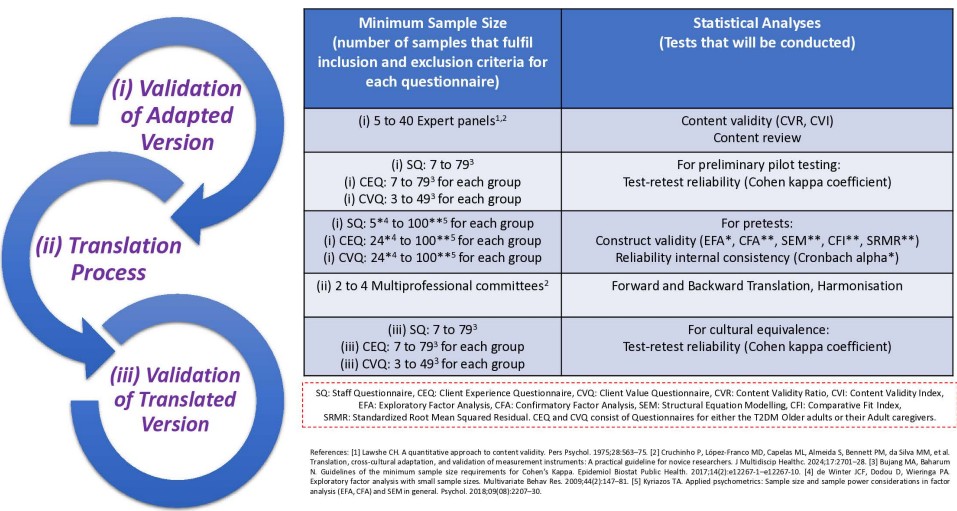

| Minimum Sample Size (number of samples that fulfil inclusion and exclusion criteria for each questionnaire) | Statistical Analyses (Tests that will be conducted) |
|---|---|
| (i) 5 to 40 Expert panels[1,2] | Content validity (CVR, CVI) Content review |
| (i) SQ: 7 to 79[3] (i) CEQ: 7 to 79[3] for each group (i) CVQ: 3 to 49[3] for each group | For preliminary pilot testing: Test-retest reliability (Cohen kappa coefficient) |
| (i) SQ: 5*[4] to 100**[5] for each group (i) CEQ: 24*[4] to 100**[5] for each group (i) CVQ: 24*[4] to 100**[5] for each group | For pretests: Construct validity (EFA*, CFA**, SEM**, CFI**, SRMR**) Reliability internal consistency (Cronbach alpha*) |
| (ii) 2 to 4 Multiprofessional committees[2] | Forward and Backward Translation, Harmonisation |
| (iii) SQ: 7 to 79[3] (iii) CEQ: 7 to 79[3] for each group (iii) CVQ: 3 to 49[3] for each group | For cultural equivalence: Test-retest reliability (Cohen kappa coefficient) |

SQ: Staff Questionnaire, CEQ: Client Experience Questionnaire, CVQ: Client Value Questionnaire, CVR: Content Validity Ratio, CVI: Content Validity Index, EFA: Exploratory Factor Analysis, CFA: Confirmatory Factor Analysis, SEM: Structural Equation Modelling, CFI: Comparative Fit Index, SRMR: Standardized Root Mean Squared Residual. CEQ and CVQ consist of Questionnaires for either the T2DM Older adults or their Adult caregivers.

References: [1] Lawshe CH. A quantitative approach to content validity. Pers Psychol. 1975;28:563–75. [2] Cruchinho P, López-Franco MD, Capelas ML, Almeida S, Bennett PM, da Silva MM, et al. Translation, cross-cultural adaptation, and validation of measurement instruments: A practical guideline for novice researchers. J Multidiscip Healthc. 2024;17:2701–28. [3] Bujang MA, Baharum N. Guidelines of the minimum sample size requirements for Cohen's Kappa. Epidemiol Biostat Public Health. 2017;14(2):e12267-1–e12267-10. [4] de Winter JCF, Dodou D, Wieringa PA. Exploratory factor analysis with small sample sizes. Multivariate Behav Res. 2009;44(2):147–81. [5] Kyriazos TA. Applied psychometrics: Sample size and sample power considerations in factor analysis (EFA, CFA) and SEM in general. Psychol. 2018;09(08):2207–30.

(i) Validation of Adapted Version

(ii) Translation Process

(iii) Validation of Translated Version

**Fig 6. Steps for conducting instruments preparation for the QUALICOPC-PCPSSP study.**

**Instruments preparation for QUALICOPC-PCPSSP study: Type (1) questionnaires for client.** The client questionnaires consist of two surveys: (i) the Client Experience (CE) Questionnaire and (ii) the Client Value (CV) Questionnaire. In the CE Questionnaire, eligible respondents will be asked to indicate their agreement with the statement by selecting 'yes' or 'no' responses [22]. The CV Questionnaire will ask eligible respondents to rate the importance of each statement in the CE survey using a scale of one (1) to four (4) [22]. The first three (3) respondents who consent to take part will fill out the CE questionnaire, while the 4th respondent will complete the CV questionnaire [17]. Clients will assess the five (5) aspects of patient-centered care, which include (i) accessibility, (ii) continuity and coordination, (iii) comprehensiveness, (iv) patient activation, and (v) doctor-patient communication [22]. According to previous research from the QUALICOPC Survey, there is compelling evidence that improving interaction and communication during appointments can strengthen the relationship between patients and PC experts [22].

**Instruments preparation for QUALICOPC-PCPSSP study: Type (2) questionnaire for staff.** The staff questionnaire evaluates workloads, considering both objective and subjective factors [23]. Objective workload is determined by the amount of work, time required, and frequency of tasks [23], while subjective workload reflects how individual workers or groups perceive their workload [23]. Workload is influenced by the composition and size of the workforce, which can impact job satisfaction [23]. Additionally, special or extra tasks assigned to specific roles can also affect staff satisfaction.

**Other instrument: Health-related Quality of Life (HRQOL) questionnaire.** The assessment of the health-related quality of life (HRQOL) in older individuals will be conducted by using the EuroQol (EQ) 5-dimensions 5-levels [EQ-5D-5L] and an EQ visual analogue scale (EQ-VAS). Before using the instrument, permission from the EuroQol website at www. EuroQol.org will be obtained. The EQ-5D-5L is a widely used generic health status measure consisting of two parts.

The first part of the questionnaire assesses health across five dimensions (mobility, self-care, usual activities, pain/discomfort, anxiety/depression), with each dimension having five levels of response (no problems, slight problems, moderate problems, severe problems, extreme problems/unable to) [24]. This comprehensive evaluation can help create a detailed health profile [24]. The second section of the questionnaire involves an EQ-VAS, where the patient assesses their perceived health on a scale from 0 (worst imaginable health) to 100 (the best imaginable health) [24]. The EQ-5D-5L questionnaire is straightforward and can be completed rapidly, with patient guidance provided within the questionnaire [24].

**Sample size.** The determination of the sample size for the QUALICOPC-PCPSSP study on clients (T2DM older adults or their caregivers) using the same formula as a cohort study, by comparing two prevalence rates, as the data was collected through an analytical cross-sectional study using questionnaires [25]. An online calculator which can be found at https://riskcalc.org/samplesize/ is used to estimate the sample size [25], with the Malaysian population's prevalence of older adults with T2DM being 33.45% being the focus [26]. This online calculator uses the formula provided by Fleiss et al. (1980) [27] for the calculation of sample size [25]. This study utilised published data on the prevalence of older adults in Malaysia for sample size calculation due to the lack of information on the proportion or prevalence of the specific comparison groups. To ensure accuracy, the sample size was calculated to be 39 in each group (2-sided significance level set at 0.05, power of 0.8, and the sample size ratio is 1). In this study, we have three (3) different groups which are 'DMTAC', 'Pharmacy VAS', and 'not enrolled with either DMTAC or pharmacy VAS'. Thus, the total sample size should be 117. With a 10% increase to account for potential dropouts, non-responses, and missing records, a recommended total sample size of 144 (each group has 48 samples) is sufficient for evaluating the performance in terms of client experience, and 48 samples (each group has 16 samples), is sufficient for evaluating the performance in terms of client value of the current public PC pharmacy services system. The calculation of each group has 16 samples is by the assumption that the sample proportion distribution approximates a normal distribution with the formula of $np \geq 5$ and $n(1-p) \geq 5$, in which `p` is a fixed value of the population proportion and `n` is the size of the random sample [28].

The QUALICOPC-PCPSSP study on pharmacy staff will involve all pharmacy personnel at the designated health clinic who meet specific criteria for inclusion and exclusion.

**Inclusion and exclusion criteria.** Clients which include patients aged 60 years or older who have been diagnosed with T2DM and/or are currently taking metformin as part of their treatment, have previously received medications during a DMTAC or Pharmacy VAS appointment (if enrolled in DMTAC or Pharmacy VAS only), have given informed consent and who can understand English and/or Malay, have an experience of at least once took medication from the public PC pharmacy as well as have expressed a desire to participate will be eligible to participate in this study. The adult caregivers aged 18 years and above of these patient criteria are also considered as clients. Patients who are under the age of 60, have been diagnosed with Type 1 diabetes, are currently using acute medications, are enrolled in DMTAC and Pharmacy VAS at the same time, or are participating in other research studies will not be eligible to participate in this study. The caregiver with these patient criteria will be excluded from this study.

Pharmacy personnel who have at least three (3) months of experience before the study enrolment, have given their informed consent, and are presently employed at the specified public health clinic will be eligible to participate in this study. Those with less than three (3) months of experience before study enrolment and/or not currently working at the designated public health clinic will not be included in this study.

Discontinuation criteria include: (i) When the research subject requests to withdraw from participating in the research or withdraws his or her consent, (ii) When the entire study is discontinued, (iii) When the principal investigator (PI) or co-investigator deems it appropriate to discontinue the research for other reasons, and/or (iv) If there are other discontinuation criteria, such as discontinuation due to undesirable events such as side effects, be described.

**Data collection.** An email or letter has been sent to the Pharmaceutical Services Division, MOH, Malaysia, to notify them that the data will be collected at public PC pharmacies in Negeri Sembilan, Malaysia for the validation study with focusing on Port Dickson Public Health Clinic for the actual study within a minimum of two (2) months. The data collection for the actual study will be conducted from March to April 2025. Approval from study sites has been obtained before the start of any study-related activities.

**Recruiting process.** The clients' surveys will be filled out in the PC pharmacy waiting area following pharmacy services. The study will include the first three (3) clients who agree to take part in the experience survey, while the 4th client will complete the values survey. Because of our resource constraints, our small sample size, not nationally collected data, and the data collection will be conducted in one (1) PC, we decided on the ratio of one (1) client for value and three (3) clients for experience questionnaires. Clients and staff will be asked for their informed consent. If respondents are not proficient in English, the Malay versions of the questionnaire will be provided. The methods for approaching clients are detailed in Table 2. Subjects will answer the questionnaire via the hard copy documents. The client's questionnaire will be printed on an A3 paper with a font size of at least 16, conversely, the staff's questionnaire will be printed on an A4 paper with a font size of at least 12. The eligible staff that meets the inclusion criteria will fill out the staff surveys in the PC pharmacy officer's room upon workdays. The researcher has set time limits of 20 minutes for client questionnaires and 30 minutes for staff questionnaire, following the protocol of a pilot study conducted in a previous international research project [12].

**Monitoring procedures.** The PI will oversee the research's advancement and manage any potential issues to ensure adherence to the protocol. All data entries will be thoroughly checked before analysis, and the research team will review the results and manuscript.

Reports to the head of the research institution shall be as follows:

• Once a year, the status of research implementation will be reported, and the suitability of continuing the research will be reviewed by the Medical Research Ethics Committee (REC), Juntendo University, School of Medicine.

• If there are any changes to the documents used for application review, application in advance will be made to the REC, Juntendo University, School of Medicine, and approval will be obtained.

**Table 2. Procedures on how client subjects will be reached.**

| Types of Client Subjects | Procedures on How Client Subjects Will be Reached |
|---|---|
| Receiving Traditional Counter Service (Not enrolled in DMTAC or Pharmacy VAS)<br><br>Receiving DMTAC | 1) Subjects will be selected during the screening process at the pharmacy counter.<br>2) If the subject is eligible and agrees to participate, medications will be prepared soon.<br>3) Subjects will be reached after receiving the medications.<br>4) Subjects will consent and complete the client`s questionnaire and quality of life questionnaire in the PC pharmacy waiting room after receiving medications.<br>5) The number will be recorded in the questionnaires. |
| Receiving Pharmacy VAS | Eligible subjects will be selected based on the daily Pharmacy VAS client list preparation worksheet. Subjects will be reached after receiving the medications. |
| • Drive Through a Pharmacy (FPL) | 1) Subjects will be identified at the Drive-Through Pharmacy (FPL) counter.<br>2) If the client agrees to participate, the subject will consent and complete the client`s questionnaire and quality of life questionnaire in the PC pharmacy waiting room after receiving medications.<br>3) The number will be recorded in the questionnaires. |
| • Integrated Medicine Dispensing System (SPUB)<br>• Pharmacy Appointment System (STJ) | 1) Subjects will be selected during the screening process at the pharmacy counter.<br>2) If the subject agrees to participate, the subject will consent and complete the client`s questionnaire and quality of life questionnaire in the PC pharmacy waiting room after receiving medications.<br>3) The number will be recorded in the questionnaires. |
| • Medicine Supply via Courrier or Postal Service (UMP) | 1) The client`s questionnaire and the client information sheet enclosed with the attached paid stamp and researcher`s address on a return envelope will be sent together with the medications via postage.<br>2) If the subject agrees to participate, the subject will consent and complete the client`s questionnaire and quality of life questionnaire at home.<br>3) The number will be recorded in the questionnaires upon receiving the feedback. |
| • Medication Locker | 1) Subjects will be selected during the collection date.<br>2) If the subject agrees to participate, the subject will consent and complete the client`s questionnaire and quality of life questionnaire in the pharmacy waiting room after receiving medications.<br>3) The number will be recorded in the questionnaires. |

• At the end of the research (including the case of cancellation or interruption), a report to the director of the research institution will be prepared.

**Data preparation for analysis: QUALICOPC-PCPSSP Survey.** All statistical analyses will be conducted using the Statistical Package for the Social Sciences (SPSS). The significance level is set at a p-value of less than 0.05.

All data will be presented as frequencies and percentages. Normality testing will be conducted on all collected data. The independent t-tests or Mann-Whitney tests will be conducted for continuous dependent variables and Pearson chi-square tests or Fisher exact tests will be conducted for categorical dependent variables of the comparison groups to assess differences between the two groups (i.e., DMTAC and Pharmacy VAS). Simple logistic regression will be used for two independent categorical variables with more than two dependent categorical variables.

Data from client questionnaires will be tabulated into various categories including (i) client demographics (for CE and CV questionnaires), (ii) client health status, frequency of pharmacy visit, and reason for current visit, (iii) client assessment of pharmacy accessibility, (iv) CE assessment (positive responses) by PC pharmacy domain, (v) CV survey responses by quality domain – the proportion of clients who ranked as 'very important', (vi) patient health-related QOL EQ-5D-5L dimensions, (vii) patient health-related EQ-VAS, (viii) patient health state index score, (ix) factors associated with DMTAC enrolment status, and (x) factors associated with Pharmacy VAS enrolment status. Comparisons for hypothesis testing will be made between DMTAC enrolment status groups ('yes' or 'not enrolled in DMTAC or Pharmacy VAS') and Pharmacy VAS enrolment status groups ('yes' or 'not enrolled in Pharmacy VAS'). Client responses to the value survey will be categorised as either `very important` (dichotomous of `very important` and `important`) or `not important` (dichotomous of

`not important` and `somewhat important`). Factors associated with the comparison groups will be analysed using simple and multiple logistic regression. The potential confounders will be identified.

The staff questionnaire data will be tabulated into four categories: (i) staff demographics, (ii) pharmacy characteristics (practice workforce composition and size), (iii) staff workload, and (iv) staff work satisfaction. Comparisons for hypothesis testing will be made on types of pharmacy occupation, including (i) pharmacist as well as (ii) pharmacy assistant and others. The study will also test the hypothesis that staff workload influences staff job satisfaction. The response options for staff work satisfaction variables will be grouped into 'agree' (dichotomous of 'strongly agree' and 'agree') and 'disagree' (dichotomous of 'disagree' and 'strongly disagree'). The composition and size of the workforce will be described.

**Data Preparation for Quality of Life (QOL) analysis.** A health profile for the study population will be created using the EQ-5D-5L questionnaire. Summary statistics will be calculated, including the number of patients and the proportions of categorical responses for the five EQ-5D-5L dimensions. A health state index score will be calculated from individual health profiles using the available value set of the Malaysian population [29]. The EQ-VAS score and health state index will be summarised using the mean and standard deviation (SD), or median. Minimum and maximum scores will be provided for DMTAC and Pharmacy VAS enrolment status groups. The notable contrast between these groups will be reported.

**Analysis plan.** The analysis plan may focus on three (3) main themes: (i) client experience of care processes (including accessibility, continuity and coordination, communication and patient-centered care, comprehensiveness and patient activation of PC pharmacy services), (ii) client value (as an indicator of the quality of PC pharmacy), and (iii) equity (related to access and the quality of PC pharmacy).

Sensitivity analyses will be conducted for this current study, which will involve two (2) different approaches to analyse the questionnaire data: (i) analysing only complete cases and (ii) imputing the missing data using single or multiple imputation methods, then redoing the analysis.

**Ethical considerations.** This research will comply with the Ethical Guidelines for Medical and Health Research Involving Human Subjects, Japan [30] and the ethical principles outlined in the 1964 Declaration of Helsinki and the International Conference on Harmonisation Guidelines for Good Clinical Practice. The study has been registered with the National Medical Research Register (NMRR), in Malaysia (NMRR ID: NMRR ID-23–03619-RYF) and has received ethics approval from the Medical Research & Ethics Committee (MREC), MOH, Malaysia dated 28th February 2024 (Reference Number: 23–03619-RYF(1) and 23–03619-RYF(2)) as well as the REC, School of Medicine of the Juntendo University in Japan dated 23rd April 2024 (Reference Number: E24-0029-M01). This research has also obtained approval from the Ministry of Economy, Malaysia dated 22nd April 2024. Study site approval from the Negeri Sembilan State Health Department has been obtained on 25th March 2024. Other relevant approvals have been received before the start of any study-related activities. All study-related activities only begin after obtaining ethics and study site approval.

**Informed consent process.** The informed consent process is an important part of this study. This process is conducted by the investigators. Before participating, all study participants are provided with a participant information sheet (PIS) approved by the MREC, Malaysia that explains the study, provides contact information for any questions, and outlines how their survey responses will be used and handled. Participation in the research is voluntary, and participants have the right to discontinue at any time without affecting their patient or work rights under the MOH. Participants will be asked to sign a consent form and date it to indicate their agreement to participate. Participation in this study requires written consent from all subjects.

**Handling of personal information.** In this study, no personal identification data will be gathered. Respondents' answers are kept anonymous, and all data will be stored in a separate offline document. The information collected only be used to meet the study's objectives and stored following ethical standards, for at least five (5) years. The study data will be securely stored offline in a password-protected electronic device, and any physical copies will be kept in a locked cabinet at the Department of Global Health Research, Graduate School of Medicine, Juntendo University, Tokyo, Japan.

The password and key access will be limited to the PI and one designated co-investigator. Once the study is completed, all other co-investigators must delete the study data from their electronic devices. This protocol will comply with all related data-sharing policies.

- Information will be provided by the following method, with names, etc. removed, so individuals cannot be identified. To do: [Main delivery method]/Direct delivery/Mail/home delivery/Electronic delivery]

- Provided to the Department of Global Health Research, Graduate School of Medicine, Juntendo University, Tokyo, Japan

- Provider: (1) Negeri Sembilan State Health Department, MOH, Malaysia (2) Pharmaceutical Services Division, Negeri Sembilan State Health Department, MOH, Malaysia.

- Information will be sent to the country of Japan.

- Kindly refer to the Personal Information Protection Commission's web page for information on the personal information protection system in the country. (URL: https://www.ppc.go.jp/personalinfo/legal/kaiseihogohou/#gaikoku)

- In addition, the measures taken to protect personal information are by referring the study protocol to the MREC, Malaysia.

**Predicted risks and benefits of the respondents.** Participation in this study will not impact the patient's current treatment and staff beneficiaries at the public health clinic, and there are no associated risks. The respondents can choose not to answer any questions that make them uncomfortable. The information from this study will help identify the strengths and weaknesses of the public PC pharmacy services system in a Malaysian setting, particularly DMTAC and pharmacy VAS, as well as job satisfaction among pharmacy staffs and initiate a discussion for health policy ideas. Participants will not be compensated for taking part in this study.

## Discussion

This QUALICOPC-PCPSSP project will provide important foundational data to support the international organizations such as European Union (EU), and the WHO generally, and specifically the Malaysian national government. This project will offer valuable insights into the organization and establishment of PC pharmacies, which can enhance the overall performance of the health system and contribute to more effective health policy discussions.

However, the causal relationship cannot be guaranteed due to the cross-sectional nature of this study. To address potential confounding factors, the researcher will employ stratification to ensure that any confounders are evenly distributed within each stratum. The experience questionnaire will be completed by the first three (3) clients, while the 4th client will fill out the value questionnaire. Previous study designs that have been used in numerous earlier international QUALICOPC studies include a large number sample size that collected national data [22] with the first nine (9) respondents answering the patient experience questionnaire and the 10th respondent answering the patient value questionnaire [17,22].

## Conclusion

This research will produce validated bilingual versions of the QUALICOPC-PCPSSP Survey instruments in English and Malay. The focus will be on assessing quality, equity, and costs. These instruments will be used to evaluate the performance of public PC pharmacy services in a Malaysian setting, concerning older adults with T2DM. Additionally, the study will also assess the job satisfaction of the pharmacy staff working in the designated public PC pharmacy setting.

This research will also evaluate the current satisfaction levels of clients who are using public PC pharmacy services (primarily DMTAC and pharmacy VAS) in Malaysia. Additionally, it will examine the QOL of older adults (aged 60 and

above) with T2DM. The study will gather baseline data on workload and staff job satisfaction in the pharmacy profession and explore how staff workload impacts job satisfaction within a public PC pharmacy services system in Malaysia.

## Means to disclose information on research

Approval from the Director General of Health, Malaysia will be obtained before publication through the National Institutes of Health (NIH), Malaysia. No personal information will be revealed when the study findings are published. All investigators are aware of the confidentiality of research-related information and should only share it with those directly involved in the research.

The findings of the research will be shared at a scholarly conference and published in a specialised journal relevant to the scientific discipline. Only statistically analysed results will be included in the publication. The identity of the participants will not be disclosed without their explicit consent when presenting or publishing the study results. No personal information will be revealed.

## Supporting information

**S1 Table. Steps for successful implementation of value-added pharmacist services at the primary care pharmacy system (structure) level.** [a]Adapted from Desselle et. al. (2019). [b]Based on 5W1H framework; need to be done whenever appropriate.
(DOC)

**S1 File. Inclusivity-in-global-research-questionnaire.**
(DOCX)

## Acknowledgments

The authors express their gratitude to the Director General of Health, Malaysia for granting permission to publish this paper. They also thank the Department of Global Health Research, Graduate School of Medicine, Juntendo University, Tokyo, Japan, and the Ministry of Health (MOH) in Malaysia in this project. Additionally, they appreciate the collaboration with the Pharmaceutical Services Division, MOH, Malaysia, and Kulliyyah of Pharmacy, IIUM, Kuantan Campus, Malaysia, for their involvement in this research.

## Author contributions

**Conceptualization:** Nurain Suleiman, Ishtiaq Ahmad.

**Formal analysis:** Nurain Suleiman.

**Investigation:** Nurain Suleiman.

**Methodology:** Nurain Suleiman, Ishtiaq Ahmad.

**Supervision:** Ishtiaq Ahmad, Motoyuki Yuasa.

**Writing – original draft:** Nurain Suleiman.

**Writing – review & editing:** Ishtiaq Ahmad, Motoyuki Yuasa.

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
