## [Decision Letter · Decision Letter 0]

PONE-D-24-29108Protocol for A QUALICOPC Study in a Malaysian Setting: Primary Care Pharmacy Services System Performance and T2DM Older Adults’ Quality of Life EvaluationPLOS ONE

Dear Dr. Ahmad,

Thank you for submitting your manuscript to PLOS ONE. After careful consideration, we feel that it has merit but does not fully meet PLOS ONE’s publication criteria as it currently stands. Therefore, we invite you to submit a revised version of the manuscript that addresses the points raised during the review process.

We look forward to receiving your revised manuscript.

Kind regards,

Samuel Chuo Yew Ting, BPharm, MPA, MMedSciPH, Ph.D

Academic Editor

PLOS ONE

Journal Requirements:

Reviewers' comments:

Reviewer's Responses to Questions

**Comments to the Author**

1. Does the manuscript provide a valid rationale for the proposed study, with clearly identified and justified research questions?

Reviewer #1: No

Reviewer #2: Yes

2. Is the protocol technically sound and planned in a manner that will lead to a meaningful outcome and allow testing the stated hypotheses?

Reviewer #1: No

Reviewer #2: Yes

3. Is the methodology feasible and described in sufficient detail to allow the work to be replicable?

Reviewer #1: No

Reviewer #2: Yes

4. Have the authors described where all data underlying the findings will be made available when the study is complete?

Reviewer #1: Yes

Reviewer #2: Yes

5. Is the manuscript presented in an intelligible fashion and written in standard English?

Reviewer #1: No

Reviewer #2: Yes

6. Review Comments to the Author

You may also provide optional suggestions and comments to authors that they might find helpful in planning their study.

Reviewer #1: The article is confusing, and its objectives, methods, and what will be conducted are not clearly stated. It needs to be thoroughly revised or rewritten.

I do not clearly understand how patient quality of life, as assessed by the EQ-5D scale, can be affected by the effectiveness of primary care (PC) pharmacies. It would be necessary to collect numerous other variables, such as patient comorbidities, which have a major impact on quality of life, to study such an association after adjustment. Furthermore, quality of life assessment is a secondary objective and not the main focus of the study. The title is unclear and should be revised, for instance: Primary Care Pharmacy Services System Performance for T2DM Older Adults: Protocol for a Cross-Sectional Study.

Are the various types of services listed in Table 1 currently implemented by pharmacies in Malaysia? If not, the authors should specify this.

Sample size: The calculation provided (which cannot be verified due to the lack of details regarding the desired precision in estimating diabetes prevalence) is irrelevant to this study. This is not a prevalence study. There is no need for a sample size calculation unless there is a specific hypothesis related to the primary endpoint.

Methods: The article would benefit from greater clarity regarding the study hypothesis, the primary objective of the study (with the associated endpoint), and the secondary objectives (with their corresponding endpoints).

Inclusion criteria: Why do the authors include the criterion "patients and/or have a spouse involved"? Similarly, participation in another study should not be considered an exclusion criterion.

What is the difference between the experience survey and the values survey?

The questionnaires for patients and pharmacists should be better described in a dedicated section or paragraph, including the number of questions, the different items evaluated, etc.

Reviewer #2: No concern on research or publication ethics

Just wondering if staff satisfaction survey includes workplace culture/environment

Moving forward, can consider covering wider range of population , example all elderly (GMTAC) as prevalance of elderly in Malaysia is more prevalence than DM in elderly. Malaysia is moving toward aged nation soon.

7. PLOS authors have the option to publish the peer review history of their article (what does this mean? ). If published, this will include your full peer review and any attached files.

**Do you want your identity to be public for this peer review?** For information about this choice, including consent withdrawal, please see our Privacy Policy .

Reviewer #1: No

Reviewer #2: No

---

## [Author Response · Author response to Decision Letter 1]

10 Feb 2025

Reviewer #1: The article is confusing, and its objectives, methods, and what will be conducted are not clearly stated. It needs to be thoroughly revised or rewritten.

Author`s feedback:

The objectives are added to the protocol as below:

‘This paper sought to conduct a more in-depth exploration of the quality of public PC pharmacy services in Malaysia as assessed by the experience of T2DM older adults or their caregivers and pharmacy staff, besides exploring some aspects of healthcare delivery that clients (could be either patients or their caregivers) value. Comparisons are also will be made between two types of major services provided by the public PC pharmacy (i.e. pharmaceutical care a.k.a. DMTAC and pharmacy VAS) as we hypothesize experiences will differ across these patient sub-groups. The secondary objective includes assessing the QOL of older adults with T2DM who are the respondents in this study and receiving the main pharmacy services in Malaysia's public PC pharmacy system (pharmaceutical care or pharmacy VAS) which is potentially useful in both the clinical encounter and quality improvement. Finally, this paper will provide possible solutions to inform healthcare reform activities moving forward‘.

These sentences also are added to the method section for clarity:

`The author highlights the lack of patient HRQOL and economic benefits evaluation of pharmacists’ roles in providing clinical services for diabetes care in a Malaysian public PC pharmacy. Furthermore, the author addressed that public PC pharmacies in a Malaysian setting are in the strategic planning in which a pharmacy has successfully implemented value-based services and has infrastructure and processes in place for sustainability, thus suggesting partnering with providers for diabetes management services. This strategy may be evaluated by clinical and humanistic outcomes [20]. Desselle et. al. (2019) emphasized that when clinical and humanistic outcomes are achieved, financial outcomes also align with them [20]. Thus, the author uses the QOL questionnaire to measure clinical outcomes and QUALICOPC questionnaires to measure human satisfaction or humanistic outcomes‘.

I do not clearly understand how patient quality of life, as assessed by the EQ-5D scale, can be affected by the effectiveness of primary care (PC) pharmacies. It would be necessary to collect numerous other variables, such as patient comorbidities, which have a major impact on quality of life, to study such an association after adjustment. Furthermore, quality of life assessment is a secondary objective and not the main focus of the study. The title is unclear and should be revised, for instance: Primary Care Pharmacy Services System Performance for T2DM Older Adults: Protocol for a Cross-Sectional Study.

Author`s feedback:

I added these sentences to the protocol:

`In a Malaysian public PC pharmacy setting, the main services offered for older adult diabetes patients include pharmaceutical care and Pharmacy Value Added Services (VAS). The offered pharmaceutical care is recognised as Diabetes Medication Therapy Adherence Clinics (DMTAC). Meanwhile, some examples of Pharmacy VAS include Drive-through pharmacy, Medicine by Post, SMS Take&Go, Appointment Card, and medicine locker, commonly referred to as Medibox [21]‘.

`The author highlights the lack of patient HRQOL and economic benefits evaluation of pharmacists roles in providing clinical services for diabetes care in a Malaysian public PC pharmacy. Furthermore, the author addressed that public PC pharmacies in a Malaysian setting are in the strategic planning in which a pharmacy has successfully implemented value-based services and has infrastructure and processes in place for sustainability, thus suggesting partnering with providers for diabetes management services. This strategy may be evaluated by clinical and humanistic outcomes [20]. Desselle et. al. (2019) emphasized that when clinical and humanistic outcomes are achieved, financial outcomes also align with them [20]. Thus, the author uses the QOL questionnaire to measure clinical outcomes and QUALICOPC questionnaires to measure human satisfaction or humanistic outcomes‘.

Additional explanation from the author:

The EQ-5D-5L questionnaire may be used for clinical settings. Higginson et. al., 2001 stated that using QOL measures in clinical practice ensures that treatment and evaluations focus on the patient rather than the disease. The measures are potentially useful in both the clinical encounter and quality improvement.

Furthermore, Marten et al.'s review of the literature in 2022 indicates that the EQ-5D-3L still has good feasibility properties and is, hence, highly applicable to older respondents.

Patients ' comorbidities will not be included since the data will be collected at the pharmacy department. Only data on the types of medications patients receive will be collected and added to the questionnaires.

The title remains the same because this is a QUALICOPC study. This study uses the adapted QUALICOPC questionnaires which have been used by more than 30 countries worldwide, and then validated to suit the Malaysian setting. Kindly refer to Fig 4. regarding the mechanism of inter-relations between the different levels and their features which has been adapted from Schäfer et. al. (2011), a QUALICOPC study. This study also evaluates older adults' QOL due to the reasons stated above, thus QOL should be mentioned in the title.

Table 1

Are the various types of services listed in Table 1 currently implemented by pharmacies in Malaysia? If not, the authors should specify this.

Author`s feedback:

I added these sentences in the protocol:

In a Malaysian public PC pharmacy setting, the main services offered for older adult diabetes patients include pharmaceutical care and Pharmacy Value Added Services (VAS). The offered pharmaceutical care is recognised as Diabetes Medication Therapy Adherence Clinics (DMTAC). Meanwhile, some examples of Pharmacy VAS include Drive-through pharmacy, Medicine by Post, SMS Take&Go, Appointment Card, and medicine locker, commonly referred to as Medibox [21].

Sample size: The calculation provided (which cannot be verified due to the lack of details regarding the desired precision in estimating diabetes prevalence) is irrelevant to this study. This is not a prevalence study. There is no need for a sample size calculation unless there is a specific hypothesis related to the primary endpoint.

Author`s feedback:

As mentioned in the protocol, comparisons for hypothesis testing will be made with the primary endpoint which includes the QUALICOPC survey data that collects client and staff responses (Kindly refer to Fig 1). Factors associated with the comparison groups will be analysed using simple and multiple logistic regression. Thus, it is rational for the sample size of the client responses should be calculated.

I revised the protocol:

`The determination of the sample size for the QUALICOPC study on clients (T2DM older adults or their caregivers) using the same formula as a cohort study, by comparing two prevalence rates, as the data was collected through an analytical cross-sectional study using questionnaires [25]. An online calculator which can be found at https://riskcalc.org/samplesize/ is used to estimate the sample size [25], with the Malaysian population's prevalence of older adults with T2DM being 33.45% being the focus [26]. This online calculator uses the formula provided by Fleiss et. al. (1980) [27] for the calculation of sample size [25]. This study utilized published data on the prevalence of older adults in Malaysia for sample size calculation due to the lack of information on the proportion or prevalence of the specific comparison groups. To ensure accuracy, the sample size was calculated to be 39 in each group (2-sided significance level set at 0.05, power of 0.8, and the sample size ratio is 1). In this study, we have three (3) different groups which are `DMTAC`, `pharmacy VAS`, and `not enrolled with either DMTAC or pharmacy VAS`. Thus, the total sample size should be 117. With a 10% increase to account for potential dropouts, non-responses, and missing records, a recommended total sample size of 144 (each group has 48 samples) is sufficient for evaluating the performance in terms of client experience, and 48 samples (each group has 16 samples), is sufficient for evaluating the performance in terms of client value of the current public PC pharmacy services system. The calculation of each group has 16 samples is by the assumption that the sample proportion distribution approximates a normal distribution with the formula of np≥5 and n(1-p)≥5, in which `p` is a fixed value of the population proportion and `n` is the size of the random sample [28]`.

The calculation from the online calculator is as follows:

Methods: The article would benefit from greater clarity regarding the study hypothesis, the primary objective of the study (with the associated endpoint), and the secondary objectives (with their corresponding endpoints).

Author`s feedback:

The hypothesis was already in the protocol as below:

‘Data gathered from the system (structure), service provision, and user levels can provide valuable insights into the strengths and areas for improvement of pharmacy services, specifically DMTAC and pharmacy VAS, within a public PC pharmacy system in Malaysia. Kindly refer to Fig 1 for the questionnaire hypotheses for both patients and pharmacy staff`.

The objectives are added to the protocol as below:

‘This paper sought to conduct a more in-depth exploration of the quality of public PC pharmacy services in Malaysia as assessed by the experience of T2DM older adults or their caregivers and pharmacy staff, besides exploring some aspects of healthcare delivery that clients (could be either patients or their caregivers) value. Comparisons are also will be made between two types of major services provided by the public PC pharmacy (i.e. pharmaceutical care a.k.a. DMTAC and pharmacy VAS) as we hypothesize experiences will differ across these patient sub-groups. The secondary objective includes assessing the QOL of older adults with T2DM who are the respondents in this study and receiving the main pharmacy services in Malaysia's public PC pharmacy system (pharmaceutical care or pharmacy VAS) which is potentially useful in both the clinical encounter and quality improvement. Finally, this paper will provide possible solutions to inform healthcare reform activities moving forward‘.

Inclusion criteria: Why do the authors include the criterion "patients and/or have a spouse involved"? Similarly, participation in another study should not be considered an exclusion criterion.

Author`s feedback:

Most T2DM older adults in Malaysia setting will come to the public PC pharmacy with their spouse and/or caregiver if they are using a wheelchair. This type of population usually will come to primary care with their caregivers or only the caregivers involved during the follow-up of the medication collection. The sentences also have been updated.

`Clients which include patients aged 60 years or older who have been diagnosed with T2DM and/or are currently taking metformin as part of their treatment, have previously received medications during a DMTAC or Pharmacy VAS appointment (if enrolled in DMTAC or Pharmacy VAS only), have given informed consent and who can understand English and/or Malay, have an experience of at least once took medication from the public PC pharmacy as well as have expressed a desire to participate will be eligible to participate in this study. The adult caregivers aged 18 years and above of this patient criteria are also considered clients. Patients who are under the age of 60, have been diagnosed with Type 1 diabetes, are currently using acute medications, are enrolled in DMTAC and pharmacy VAS at the same time, or are participating in other research studies will not be eligible to participate in this study. The caregiver with this patient criteria will be excluded from this study`.

Additional explanation from the author:

To avoid confusion and misleading or contamination of the results, participation in any other study will be excluded.

What is the difference between the experience survey and the values survey?

Author`s feedback:

Client Experience is related to the Primary Care Pharmacy Domain, in which patients rated either `yes` or `no` based on their experience. In contrast, Client Value is related to Quality Domain, in which patients rated either `very important`, `important`, `not important`, or `somewhat important`. Both questionnaires are concerning the assessment of 5 factors (i) accessibility, (ii) continuity and coordination, (iii) communication and patient-centered care, (iv) patient activation, and (v) comprehensiveness. The respondents for client questionnaires are either T2DM older adults or their caregivers.

The questionnaires for patients and pharmacists should be better described in a dedicated section or paragraph, including the number of questions, the different items evaluated, etc.

Author`s feedback:

Figure 2 states that the questionnaire for staff will be adapted from 2 main published papers (Reference numbers 23 and 30). The number of questions will be finalised after the validation of the questionnaire is completed.

Reviewer #2: No concern on research or publication ethics

Author`s feedback:

It is mentioned in the ‘Means to Disclose Information on Research‘.

Just wondering if the staff satisfaction survey includes workplace culture/environment

Author`s feedback:

Figure 2 states that the questionnaire for staff will be adapted from 2 main published papers (Reference numbers 23 and 30). The questions will be finalised after the validation of the questionnaire is completed.

Moving forward, can consider covering a wider range of populations, for example, all elderly (GMTAC) as the prevalence of elderly in Malaysia is more prevalent than DM in the elderly. Malaysia is moving toward an aged nation soon.

Author`s feedback:

This study is focused on the main services (i.e. pharmaceutical care a.k.a. DMTAC in the Malaysian setting, and pharmacy VAS) provided by the public PC pharmacy to diabetes patients. The other pharmaceutical care service may not yet be provided by the public PC pharmacy.

7. PLOS authors have the option to publish the peer-review history of their article (what does this mean?). If published, this will include your full peer review and any attached files.

Note: Highlighted in yellow are the changes that have been made in protocol

---

## [Editor Report · Decision Letter 1]

PONE-D-24-29108R1Protocol for A QUALICOPC Study in a Malaysian Setting: Primary Care Pharmacy Services System Performance and T2DM Older Adults’ Quality of Life EvaluationPLOS ONE

Dear Dr. Ahmad,

Thank you for submitting your manuscript to PLOS ONE. After careful consideration, we feel that it has merit but does not fully meet PLOS ONE’s publication criteria as it currently stands. Therefore, we invite you to submit a revised version of the manuscript that addresses the points raised during the review process.

We look forward to receiving your revised manuscript.

Kind regards,

Samuel Chuo Yew Ting, BPharm, MPA, MMedSciPH, Ph.D

Academic Editor

PLOS ONE

**Journal Requirements:**

**Additional Editor Comments:**

Please address the comments from reviewers promptly.

---

## [Author Response · Author response to Decision Letter 2]

26 Mar 2025

(i) A rebuttal letter that responds to each point raised by the academic editor and reviewer(s). You should upload this letter as a separate file labeled 'Response to Reviewers' - as attachment

(ii) A marked-up copy of your manuscript that highlights changes made to the original version. You should upload this as a separate file labeled 'Revised Manuscript with Track Changes' - as attachment.

(iii) An unmarked version of your revised paper without tracked changes. You should upload this as a separate file labeled 'Manuscript'.

(iv) If applicable, we recommend that you deposit your laboratory protocols in protocols.io to enhance the reproducibility of your results. Protocols.io assigns your protocol its own identifier (DOI) so that it can be cited independently in the future. For instructions see: https://journals.plos.org/plosone/s/submission-guidelines#loc-laboratory-protocols. Additionally, PLOS ONE offers an option for publishing peer-reviewed Lab Protocol articles, which describe protocols hosted on protocols.io. Read more information on sharing protocols at https://plos.org/protocols?utm_medium=editorial-email&utm_source=authorletters&utm_campaign=protocols - we do not apply for this because we are preparing a study protocol, not a lab protocol

Journal Requirements:

(v) Please review your reference list to ensure that it is complete and correct. If you have cited papers that have been retracted, please include the rationale for doing so in the manuscript text, or remove these references and replace them with relevant current references. Any changes to the reference list should be mentioned in the rebuttal letter that accompanies your revised manuscript. If you need to cite a retracted article, indicate the article’s retracted status in the References list and also include a citation and full reference for the retraction notice - updated reference list

(vi) Please address the comments from reviewers promptly - the comments were addressed more properly.

---

## [Editor Report · Decision Letter 2]

Protocol for A QUALICOPC Study in a Malaysian Setting: Primary Care Pharmacy Services System Performance and T2DM Older Adults’ Quality of Life Evaluation

PONE-D-24-29108R2

Dear Dr. Ahmad,

We’re pleased to inform you that your manuscript has been judged scientifically suitable for publication and will be formally accepted for publication once it meets all outstanding technical requirements.

Kind regards,

Samuel Chuo Yew Ting, BPharm, MPA, MMedSciPH, Ph.D

Academic Editor

PLOS ONE
---

## [Editor Report · Acceptance letter]

PONE-D-24-29108R2

PLOS ONE

Dear Dr. Ahmad,

I'm pleased to inform you that your manuscript has been deemed suitable for publication in PLOS ONE. Congratulations! Your manuscript is now being handed over to our production team.

Kind regards,

on behalf of

Dr. Samuel Chuo Yew Ting

Academic Editor

PLOS ONE